# Radiation Dose and Fluoroscopy Time of Endovascular Coil Embolization in Patients with Carotid Cavernous Fistulas

**DOI:** 10.3390/diagnostics12020531

**Published:** 2022-02-18

**Authors:** Yigit Ozpeynirci, Christoph Gregor Trumm, Robert Stahl, Thomas Liebig, Robert Forbrig

**Affiliations:** Institute of Neuroradiology, University Hospital, LMU Munich, Marchioninistr. 15, 81377 Munich, Germany; christoph.trumm@med.uni-muenchen.de (C.G.T.); robert.stahl@med.uni-muenchen.de (R.S.); thomas.liebig@med.uni-muenchen.de (T.L.); robert.forbrig@med.uni-muenchen.de (R.F.)

**Keywords:** arteriovenous fistula, cerebral angiography, radiation exposure, endovascular procedure, cerebrovascular disorders

## Abstract

Carotid cavernous fistulas (CCFs) are abnormal connections between the cavernous sinus and the internal and/or external carotid artery. Endovascular therapy is the gold standard treatment. In the current retrospective single-center study we report detailed dosimetrics of all patients with CCFs treated by endovascular coil embolization between January 2012 and August 2021. Procedural and dosimetric data were compared between direct and indirect fistulas according to Barrow et al., and different DSA protocol groups. The local diagnostic reference level (DRL) was defined as the 3rd quartile of the dose distribution. In total, thirty patients met the study criteria. The local DRL was 376.2 Gy cm^2^. The procedural dose area product (DAP) (*p* = 0.03) and the number of implanted coils (*p* = 0.02) were significantly lower in direct fistulas. The median values for fluoroscopy time (FT) (*p* = 0.08) and number of DSA acquisitions (*p* = 0.84) were not significantly different between groups. There was a significantly positive correlation between DAP and FT (*p* = 0.003). The application of a dedicated low-dose protocol yielded a 32.6% DAP reduction. In conclusion, this study provides novel DRLs for endovascular CCF treatment using detachable coils. The data presented in this work might be used to establish new specific DRLs.

## 1. Introduction

Carotid cavernous fistulas (CCFs) are abnormal arteriovenous (AV) connections between the cavernous sinus (CS) and the internal carotid artery (ICA) or dural branches of the ICA and/or external carotid artery (ECA) [1,2,3].

Exophthalmos, retroorbital pain, chemosis, ophthalmoplegia, and fast decline of visual acuity, eventually leading to permanent vision loss, are symptoms of a CCF. Furthermore, CCF-induced reflux into the cortical cerebral veins (which normally drain into the CS) may yield venous congestion and intracranial hemorrhage [2,3].

Barrow et al., introduced the most commonly used classification, which categorizes CCFs into two main groups (direct and indirect) and four subgroups (Type A–D) [1]. Direct fistulas (Type A) represent pathological AV communications between the ICA and the CS. They are usually caused by laceration of the cavernous ICA segment following a severe head trauma with or without a concomitant skull base fracture, rupture of a cavernous ICA aneurysm, genetic connective tissue disorders (e.g., Ehlers-Danlos syndrome or fibromuscular dysplasia) and/or iatrogenic injuries (e.g., surgical trauma). Direct CCFs usually require emergent treatment because of their high-flow nature, acute presentation and the possibility of rapid deterioration. In contrast, indirect fistulas (Type B–D) are characterized by uni- or bilateral connections between the dural branches of the ICA (Barrow Type B), ECA (Type C), or both arteries (Type D) and the CS. The causes of indirect CCFs are unknown, but reported predisposing factors are pregnancy, sinusitis, trauma, CS thrombosis, surgery, female gender and age > 50 years. Indirect CCFs typically yield a low-flow AV shunt with gradually progressive and subacute symptoms [1,2,3].

Treatment methods consist of microneurosurgery, stereotactic radiosurgery and endovascular embolization. In recent years, fluoroscopically-guided endovascular therapy has become the gold standard treatment of CCF, as this technique allows for higher and faster cure rates as well as lower procedure-related morbidity when compared to other methods [3,4,5,6,7,8,9]. In particular, previous studies have shown that endovascular embolization through a transvenous route with detachable coils is considered the most effective treatment of CCF [6,7,8,9]. The favorite route to the CS described in the literature is through the inferior petrosal sinus [6]. In a recent meta-analysis consisting of 22 studies reporting 1066 procedures in 1043 patients with indirect CCFs, Alexandre et al., showed that transvenous coiling was the most common treatment approach (712/1066, 57.8%), yielding both high radiological and clinical success as well as low complication rates [10].

However, as for every procedure involving radiation exposure to both the patient and the physician, diagnostic reference levels (DRL) are needed as a quality improvement tool and to maintain diagnostic and therapeutic standards. Regarding neuroendovascular procedures, the current national guidelines for radiation protection, updated in 2018 [11], define national DRLs only for mechanical thrombectomy and embolization of an intracranial aneurysm [12]. In recent publications of our research group, local DRLs for endovascular treatment of unruptured intracranial aneurysms and intracranial lateral dural arteriovenous fistulas (DAVF), as well as for diagnostic angiographies in patients with spinal DAVFs, were defined [13,14,15] in order to expand the neuroendovascular DRL collection. In the current monocentric study, detailed dosimetrics in CCF patients treated by fluoroscopically-guided coil embolization are presented and compared between patients with direct and indirect CCFs, as well as between various DSA acquisition protocols. These data might be valuable in order to introduce novel dedicated DRLs, as only a few dosimetry articles specifically dealing with the endovascular treatment of this subtype of intracranial fistulas have been published so far [16].

## 2. Materials and Methods

This retrospective single-centre study was approved by the responsible Institutional Review Board of the Ludwig-Maximilians-University Munich, Germany (project number 20-450). A retrospective data analysis from all consecutive patients who presented with a CCF treated at our institution between January 2012 and August 2021 was performed. The following inclusion and exclusion criteria were set:

Inclusion criteria:-Age ≥ 18 years-Presence of a fistula between the ICA and/or ECA and the CS-Treatment with endovascular embolization using only detachable coils

Exclusion criteria:-CCFs treated with alternative endovascular methods (e.g., detachable balloons, liquid embolic agents, covered stents, flow-diverter stents or combined techniques)-Recurrent fistulas-Fistulas located elsewhere but also treated with CS coil embolization

### 2.1. Procedure

All endovascular procedures were performed under general anesthesia using a biplane angiographic unit (Axiom Artis Zee, Siemens Healthineers, Erlangen, Germany) by six experienced neuroradiologists with between five and more than 20 years of experience in neurointerventions. In all patients, arterial and/or venous access was achieved using a transfemoral approach. The non-ionic iodinated contrast agent applied was Iomeprol 300 mg iodine/mL (Imeron, Bracco Imaging, Kontanz, Germany). Regarding patients treated by a transarterial approach, after placing a guiding catheter in the target ICA, the venous side of the fistula was catheterized with a 0.017-inch microcatheter through the tear in the ICA wall. In patients treated by a transvenous approach, the microcatheter was advanced through the inferior petrosal sinus into the fistulous CS compartment. In both methods, the fistulous CS part was filled using detachable platinum coils until the AV shunt was fully closed. The initial and final diagnostic digital subtraction angiography (DSA) acquisitions on standard anteriorposterior and lateral projections were performed using a field of view (FOV) of 32 cm and a frame rate of 1–4 frames/second (f/s). DSA acquisitions on working projections were performed using a targeted FOV of either 11 cm or 16 cm and a higher frame rate of up to 7.5 f/s. The frame rate of pulsed fluoroscopy was 7.5 f/s. With respect to the DSA acquisition type, two protocols were preset by the manufacturer as previously reported [13] and applied under discretion of the treating physicians:-Low-dose (LD): tube voltage 73 kV, pulse width 50 ms, dose 1820 μGy/pulse-Normal-dose (ND): tube voltage 73 kV, pulse width 100 ms, dose 3000 μGy/pulse

In mixed-dose (MD) cases, both DSA protocols (LD and ND) were used during the same intervention. In order to examine the impact of different DSA protocols on the radiation exposure, two groups were formed: (1) LD group and (2) ND/MD group.

### 2.2. Data Collection and Dosimetry Analysis

Patient demographics (age, sex) and procedural data (fistula type and site, access route to the fistula, immediate angiographic success of the treatment and number of detached coils) were retrospectively obtained from the medical charts. Imaging data and dose reports were retrieved from a picture archiving and communication system (syngo.imaging, Siemens Healthineers). In terms of procedural radiation exposure, the following dosimetry parameters were collected: dose area product (DAP, Gy cm^2^), fluoroscopy time (FT) and number of DSA acquisitions. The total DAP was calculated by adding the DAP of fluoroscopy and DSA acquisitions together.

### 2.3. Statistics

Data were initially tested for normality using the Shapiro-Wilk test. Continuous variables were presented as means ± standard deviation (sd), percentages and ranges. Counts and percentages were calculated to represent categorical data. The local DRL for the endovascular coil embolization of CCFs was set at the third quartile value of the DAP distribution [17].

The two-sided unpaired *t*-test was applied to compare procedural DAP, FT, and number of detached coils between direct and indirect fistulas. The Mann-Whitney U test was used to compare the number of DSA acquisitions between cases with direct and indirect fistulas. The same tests were used to compare the same variables between the LD group and the ND/MD group. Furthermore, a Spearman correlation analysis was performed to determine the relationship between procedural DAP, FT, number of DSA acquisitions and implanted coils. A linear regression analysis was made between procedural DAP and FT.

All calculations were carried out using SPSS software Version 26.0 (IBM, Armonk, New York, NY, USA). A *p*-value < 0.05 was considered statistically significant.

## 3. Results

Between January 2012 and August 2021, 30 patients with CCFs treated by endovascular means met the inclusion and exclusion criteria. Table 1 shows patient demographics and procedural data. Causes of direct fistulas were a ruptured aneurysm of the cavernous ICA segment (n = 4), severe head trauma (n = 2), and spontaneous ICA dissection (n = 1). No procedure-related complications, which would have prolonged the intervention time, were documented.

The local DRL was 376.2 Gy cm^2^. The 75th percentile of the FT distribution was 241.8 min (Table 2). 7/30 (23%) patients exceeded the local DRL. All of these patients had an indirect fistula.

Regarding the fistula type, significantly lower median values were calculated in patients with direct CCFs both for procedural DAP (*p* = 0.03) and number of implanted coils (*p* = 0.02) when compared to patients with indirect CCFs. Median values for FT (*p* = 0.08) and number of DSA acquisitions (*p* = 0.84) were not significantly different between groups (Figure 1, Table 2 and Table 3).

Pair-wise correlations of the dosimetrics (DAP, FT, and number of DSA acquisitions) and number of implanted coils showed a significantly positive correlation only between radiation dose and FT (*p* = 0.003, R^2^ = 0.3), with a DAP increase of 0.96 Gy cm^2^ per additional minute of FT according to the linear regression analysis (Figure 2).

An LD DSA protocol was used in 20/30 (68%) patients whereas an ND or MD DSA protocol was applied in 10/30 (32%) patients. Regarding radiation exposure, the mean DAP was significantly lower in the LD group when compared to the ND/MD group (274.6 versus 407.4 Gy cm^2^, *p* = 0.02). In contrast, no significant differences between groups were calculated for the mean FT (LD 186 versus ND/MD 179.5 min, *p* = 0.84), mean number of implanted coils (LD 22.4 versus ND/MD 20.3, *p* = 0.65) and mean DSA acquisitions (LD 40.7 versus ND/MD 40.8, *p* = 0.58) (Table 4).

## 4. Discussion

International advisory organizations on ionizing radiation safety emphasize the importance of the justification of patients’ exposure to radiation, as well as the need of recording the radiation dose of each examination and applying suitable DRLs [17].

A comprehensive literature search revealed no officially established or recommended DRLs for the endovascular treatment of CCFs [12,18,19,20]. In this retrospective, single-center study, detailed dosimetry data are reported from 30 CCFs endovascularly treated by coil embolization. The mean procedural DAP was 318.8 Gy cm^2^ and the average FT was 183.8 min. Local 3rd quartile values were 376.2 Gy cm^2^ for DAP (i.e., DRL) and 241.8 min for FT.

Only in a few dosimetry studies, have intracranial DAFVs been represented as a subcategory [14,21,22]. The reported results in terms of local DRLs were quite various (414 Gy cm^2^, 507.33 Gy cm^2^ and 730 Gy cm^2^). However, Forbrig et al. [14] and Opitz et al. [21] excluded CCFs, and Kien et al. [22] did not describe the inclusion criteria in detail. Regarding FT, Forbrig et al., proposed 142 min and Kien et al., 80 min.

Intracranial DAFVs represent a heterogenous group of vascular pathologies handled in different ways using various embolic materials. As a result, local DRLs suggested by multiple neurovascular centers may differ substantially, consequently disabling adequate comparison with our findings.

In the only study published on radiation exposure in endovascular management of CCFs, Opitz et al. [16] proposed a DRL of 350 Gy cm^2^, which is in line with our observations. The mean FT in our study group was yet substantially higher than in their study (183.8 vs. 61.9 min). One explanation for this finding might be a comparably higher number of technically challenging procedures in our study group due to a difficult vascular anatomy in some cases, which in turn may have required a longer time for successful catheterization of the fistula-harboring CS compartment. In this context, a longer FT alone does not necessarily imply a significantly increased radiation dose, because most of the procedural irradiation stems from DSA acquisitions but not from pulsed fluoroscopy. However, as expected, a positive correlation between FT and procedural DAP (*p* = 0.003) was found in our study group.

Direct CCFs usually develop suddenly because of a tear on the ICA wall (e.g., after aneurysm rupture), yielding high-flow AV shunts in most cases. Thus, they are commonly diagnosed earlier than indirect fistulas. The therapeutic strategy, depending on the cause, is either transarterial treatment of the aneurysm or transvenous occlusion of the fistulous CS compartment. In contrast, indirect CCFs are commonly low-flow, chronic lesions and are accompanied by thrombotic occlusion of the draining veins, resulting in a technically more challenging CS catheterization. Furthermore, in indirect CCFs the fistulous CS compartment is commonly larger when compared to direct fistulas. Accordingly, in the present study values for both the periprocedural DAP (*p* = 0.03) and number of implanted coils (*p* = 0.02) were significantly higher in patients with indirect fistulas when compared to those with direct fistulas. Moreover, each of the seven patients exceeding our local DRL level had an indirect fistula type.

There was no correlation between the amount of coils used and FT, radiation exposure or number of DSA runs. These data outline the fact that navigation of the microcatheter towards the target (i.e., fistulous CS compartment) represents the most time-consuming part of the intervention, and that the number of subsequently detached coils does not substantially affect overall procedural radiation dose. Figure 3 illustrates two patients with indirect fistulas endovascularly treated by 9 and 55 coils, respectively. DAP values were comparable between both cases (316.1 vs. 330.5 Gy cm^2^).

Regarding radiation dose optimization in the field of interventional neuroradiology, several techniques have been proposed in recent years [13,23,24,25,26]. At our institution, neurointerventionists may choose between two preset DSA protocols (LD and ND) dependent on the patient age, vascular pathology, purpose of the examination (diagnostic or therapeutic) or required image resolution (e.g., ND during superselective catheterization of the fistula point and LD during coiling of the fistula). In this study, patients in the LD group had a significantly lower radiation exposure compared to the other group (mean DAP 274.6 versus 407.4 Gy cm^2^, *p* = 0.02), without any significant difference regarding further procedural parameters such as FT, number of DSA acquisitions and number of implanted coils. The LD protocol yielded a dose reduction of 32.6%. The positive impact of this dedicated DSA protocol on radiation dose was also demonstrated in recently published papers from our institution [13,14,15].

For the determination of local DRLs with acceptable 95% confidence intervals, Miller et al., propose at least 30 studies of the same procedure [27]. Hence, the dosimetry results of our study population may substantially contribute to the literature in order to introduce novel DRLs in the field of endovascular CCF treatment. Although rare, these vascular disorders represent an important subgroup of (often time-consuming) neurointerventional procedures, thus demanding proper documentation and optimization of radiation exposure.

This study has several limitations because of the mono-centric design. Angiographies were performed using a single angiographic system from a single vendor (Siemens Healthineers). Furthermore, patients treated with liquid embolics or direct puncture of the ophthalmic vein were explicitly excluded for the sake of homogeneity, hence the provided dosimetric data cannot be generalized to the entire treatment spectrum of CCFs. However, this study still covers the majority of patients undergoing endovascular CCF treatment.

## 5. Conclusions

The present study provides novel local DRLs in the field of endovascular treatment of CCFs using detachable coils, the most established and durable therapy option. Procedural radiation exposure was significantly higher in patients with indirect fistulas when compared to direct fistulas. Regarding radiation dose optimization, the application of a dedicated LD DSA protocol allowed for a 32.6% DAP reduction.

## Figures and Tables

**Figure 1 diagnostics-12-00531-f001:**
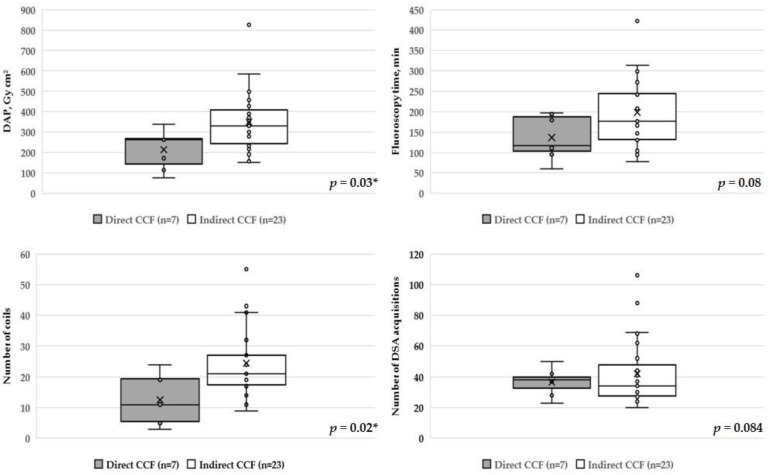
Box plots and scatter plots of total DAP, FT, number of implanted coils, and DSA acquisitions showing different types of CCFs. Significantly higher DAP (*p* = 0.03) and number of implanted coils (*p* = 0.02) in the indirect group. No significant difference in FT (*p* = 0.08) or number of DSA acquisitions (*p* = 0.84) between two groups. DAP: dose area product, FT: fluoroscopy time, DSA: digital subtraction angiography, CCF: carotid cavernous fistula * indicates statistically significant difference.

**Figure 2 diagnostics-12-00531-f002:**
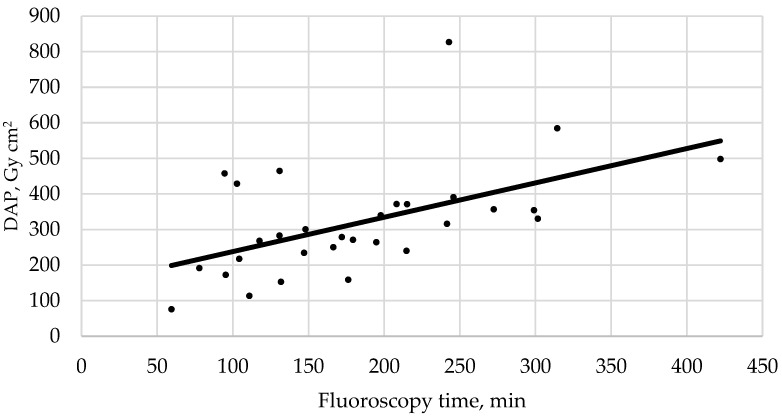
The relationship between fluoroscopy time and radiation exposure is summarized by the fitted regression line.

**Figure 3 diagnostics-12-00531-f003:**
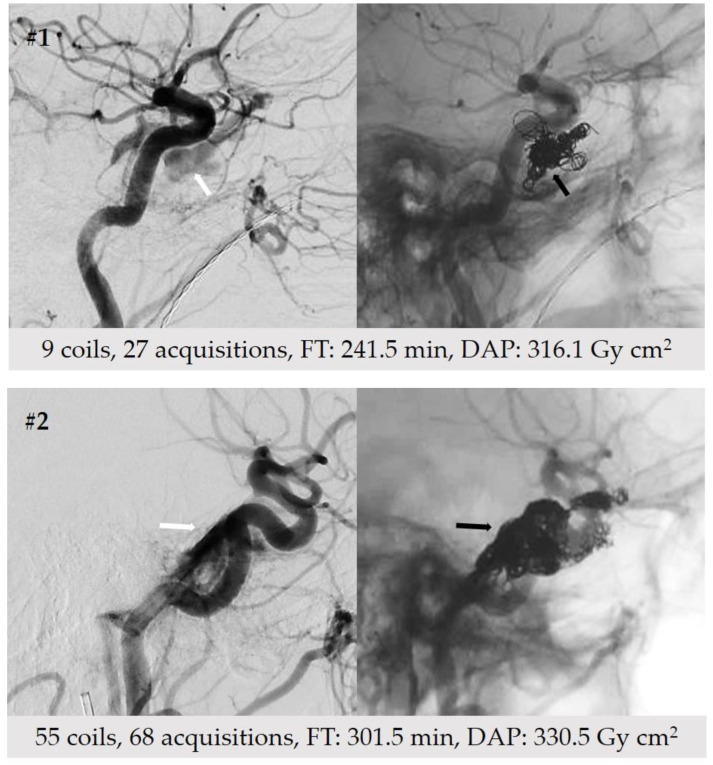
Dosimetrics of two exemplary patients with indirect fistulas. Patient #1 was endovascularly treated with the lowest (n = 9) and patient #2 with the highest number of coils (n = 55) in our study cohort. White arrows point to early filling of the cavernous sinus with contrast. Black arrows indicate implanted coils within the fistulous compartment.

**Table 1 diagnostics-12-00531-t001:** Patient demographics and procedural data.

Mean Age, Years (Range)	66 (41–85)
Female sex	23/30 (77%)
Number of CCFs	
-Direct (Type A)	7/30 (23%)
-Indirect ○ Type B ○ Type C ○ Type D	23/30 (77%) 4/30 (13%) 2/30 (7%) 17/30 (57%)
Angiographic outcome	
-Total occlusion	19/30 (63%)
-Small remnant	11/30 (37%)
Approach type	
-Transarterial	2/30 (7%)
-Transvenous	27/30 (90%)
-Combined	1/30 (3%)
Fistula site	
-Right	9/30 (30%)
-Left	8/30 (27%)
-Bilateral	13/30 (43%)

CCF: carotid cavernous fistula.

**Table 2 diagnostics-12-00531-t002:** Distribution of radiation dose and fluoroscopy time.

	DAP, Gy cm^2^			Fluoroscopy Time, Minutes		
	25th percentile	Median	75th percentile	25th percentile	Median	75th percentile
All (n = 30)	230.2	291.8	376.2	115.9	174.1	241.8
Direct (n = 7)	113.7	264.2	271.2	95.3	117.6	194.8
Indirect (n = 23)	240.2	330.5	428.8	130.9	176.2	245.8

DAP: dose area product.

**Table 3 diagnostics-12-00531-t003:** Dosimetrics and number of implanted coils.

	All CCFs (n = 30)	Direct CCFs (n = 7)	Indirect CCFs (n = 23)
Mean DAP, Gy cm^2^ (range)	318.8 ± 148.1 (75.9–826.6)	**215.1 ± 89 (75.9–339.9)**	**350.4 ± 148.1 (153–826.6)**
Median DAP, Gy cm^2^	291.8	264.2	330.5
Mean FT, min (range)	183.8 ± 81.4 (59.4–422.2)	136.4 ± 50.2 (59.4–197.7)	198.3 ± 83.6 (77.8–422.2)
Median FT, min	174.1	117.6	176.2
Mean number of DSA runs (range)	40.8 ± 19.6 (20–106)	36.6 ± 8.2 (23–50)	42 ± 21.8 (20–106)
Median number of DSA runs	35.5	38	34
Mean number of detached coils (range)	21.7 ± 11.7 (3–55)	**12.6 ± 7.8 (3–24)**	**24.5 ± 11.2 (9–55)**
Median number of detached coils	19.5	11	21

CCF: carotid cavernous fistula, DAP: dose area product, FT: fluoroscopy time. Values that are significantly different between direct and indirect CCFs are written in bold.

**Table 4 diagnostics-12-00531-t004:** Comparison of dosimetric data and number of implanted coils between treatment groups with different DSA protocols.

	All CCFs (n = 30)	Low-Dose (n = 20)	Normal- and Mixed-Dose (n = 10)	*p* Value
Mean DAP, Gy cm^2^	318.8	**274.6**	**407.4**	**0.02**
Mean FT, min	183.8	186.1	179.5	0.84
Mean number of detached coils	21.7	22.4	20.3	0.65
Mean number of DSA acquisitions	40.8	40.7	40.8	0.58

CCF: carotid cavernous fistula, DAP: dose area product, DSA, digital subtraction angiography, FT: fluoroscopy time. Values that are significantly different are written in bold.

## Data Availability

Not applicable.

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
