# Peer review of "Radiation Dose and Fluoroscopy Time of Endovascular Coil Embolization in Patients with Carotid Cavernous Fistulas"

_diagnostics, 2022, doi:10.3390/diagnostics12020531_

Round 1

Reviewer 1 Report

The author analyzed local DRL for CCFs. These data are very important for radioprotection and dose optimization in endovascular therapy. I have a few suggestions for this study.

1) Did you analyze cumulative dose at the interventional reference point? If yes, please added in paper.

2) Add the indirect CCF data to Table 2.

3) Add the individual data points to Fig. 1. Like Fig.3 of https://doi.org/10.1038/s41586-021-04234-3

4) Line 186, Show these results in a figure.

5) Line 203 and 208, summarize the present data and previous reports in table.

6) Line 211, please make discussion about why lower DAP values but a longer FT are observed in preset study.

7) What criteria do doctors use to choose between ND and LD?

Author Response

Thank you for the comments. You will find our point-by-point responses and the revised version of the text attached.

Reviewer 2 Report

The manuscript entitled "Radiation Dose and Fluoroscopy Time of Endovascular Coil Embolization in Patients with Carotid Cavernous Fistulas" shows the results of a retrospective analysis of 30 patients who underwent fluoroscopy based endovascular procedure to treat CCF.

The results and conclusions are of potential interest to the readers of this journal. Overall, the paper is well written. However, I considered that the results might be presented in a more straightforward and intelligible way. I have some remarks that might help improve the manuscript's readability.

General and specific comments:

  1. Avoid pronouns and non-scientific language such as "we believe" and "we suggested". Please use passive voice instead. (e.g. "The results might....", "This study shows..."). 
  2. Abstract. The aim of the study is not clearly stated in the abstract. Please re-write the sentence "We believe that our data....." , for example, "The data presented in this work might be used to establish new specific DRLs..." or something similar.
  3. Introduction. Between lines 47-49, a paragraph is missing linking the classification of CCFs and its treatment by fluoroscopy therapy. A brief paragraph mentioning the overall treatment options.
  4. Lines 50-51. It would be of interest to mention the proportion of CCFs coil embolization relative to the other treatment options.
  5. Line 79. Why only detachable coils?. It would be necessary to the reader to understand why only the authors include this treatment option. The reason for this inclusion of criteria must be clearly stated in the manuscript.
  6. Line 122. Please, authors must clearly explain how the DAPs were combined. Adding, averaging, etc.
  7. Lines 133-134. The redaction of this paragraph is a little confusing. I suggest to the authors re-write this paragraph. It must be clear to the reader how the statistical analysis is linked to the DRL definition (lines 135-138) and how these methods aid in fulfilling the goal of this work.
  8. Line 218. Some possible explanations for this result? Why do the authors mention that FT does not imply a significantly increased radiation dose if there is a positive correlation? Their results showed that indeed is significant (p=0.003). Please, clarify. 
  9. Lines 274-275. Since it is not clearly stated the aim of the study, the conclusions are not clear. Please, clearly state the study's goal and modify the conclusions accordingly if necessary. 

Author Response

(The authors gave the same response as above.)

Reviewer 3 Report

Interesting and well written study regarding Radiation Dose and Fluoroscopy Time of Endovascular Coil Embolization in Patients with Carotid Cavernous Fistulas. 

This aspect is of particular interest in interventional neuroradiology, and very often is underestimated by operators.

The article analyses both direct and indirect ccf. Regarding the first to the best of my knowledge the are no sisytematic review of meta-analysis analysing these concetps. Regarging indirect CCF (best known ad dural arterio-venous fistula to the carvernous sinus) I suggest you to cite and coment this meta-analysis, which is the most comprensive in the current Literature: Alexandre AM, Sturiale CL, Bartolo A, Romi A, Scerrati A, Flacco ME, D'Argento F, Scarcia L, Garignano G, Valente I, Lozupone E, Pedicelli A. Endovascular Treatment of Cavernous Sinus Dural Arteriovenous Fistulas. Institutional Series, Systematic Review and Meta-Analysis. Clin Neuroradiol. 2021 Dec 15. doi: 10.1007/s00062-021-01107-0. Epub ahead of print. PMID: 34910224.

Just minor English correction needed. 

Results and conclusions are appropriate. 

Author Response

(The authors gave the same response as above.)
